

# Dune belt restoration effectiveness assessed by UAV topographic surveys (Northern Adriatic coast, Italy)

Regine Anne Faelga [1*], Luigi Cantelli [1], Sonia Silvestri [1], Beatrice Maria Sole Giambastiani [1]

[1] Biological, Geological and Environmental Sciences Department, University of Bologna, Bologna, 40126, Italy

*Correspondence to*: Regine Anne Faelga (regineanne.faelga@studio.unibo.it)

**Abstract.** Unmanned Aerial Vehicle (UAV) monitoring surveys are used to assess a dune restoration project in the protected natural area of the Bevano River mouth in the Northern Adriatic coast (Ravenna, Italy). UAV is among the most utilized tools in coastal geomorphology studies as high-spatial and temporal resolution surveys can be carried out in an efficient and
cost-effective manner. The impact of the installed fences to dune development are assessed in terms of sand volume and vegetation cover changes over time by using a systematic data processing workflow based on Structure from Motion (SfM) photogrammetry and Geomorphic Change Detection (GCD) toolset. Accuracy assessment is performed using statistical analysis between GPS profiles and the elevation models. Results show that the dune fence proves to be effective to prevent dune erosion since significant sand accumulation is observed along the dune foot and front. Progradation of around 3-5 m of
the foredune, development of embryo dunes, decrease in stoss slope and blowout features due to increase in vegetation colonization were observed. Erosion is evident at the northern portion of the structure, which could be accounted for by the aerodynamic and morphodynamic conditions around the dune fence, the efficiency of the fence and its configuration to trap sediments. Dune fencing and limiting debris cleaning along the protected coast has been proven to be very effective against dune degradation. The GCD toolset can be a valuable tool if sources of uncertainties are well accounted for. The proposed
workflow can also aid in creating transferable guidelines to stakeholders in ICZM implementation in the Mediterranean low-lying sandy coasts.

## 1. Introduction

Coastal dunes are significant ecosystems that can provide flood protection, groundwater storage, salinization prevention, species habitat, and recreation. Their dynamics are driven by the complex interaction between the controlling winds,
vegetation, and the nearshore-beach geomorphology (Sloss et al., 2012; Lalimi et al., 2017). The highly dynamic nature, in addition to climatic and anthropogenic pressures make these landforms extremely vulnerable. To prevent further degradation, soft or limited engineering, along with Nature-Based Solutions (NBS) have been the preferred intervention solutions for coastal zones as they enable a more dynamic evolution and functioning. In Europe, coastal foredunes have been stabilized over the past century by reprofiling, planting vegetation and dune fencing, and/or beach nourishment (Matias et



al., 2005; Nordstrom and Arens, 1998; Arens et al., 2001; Ruz and Anthony, 2008; De Vriend and Van Koningsveld, 2012; Laporte-Fauret et al., 2021).

Advances in coastal dune geomorphology studies have been evident through the availability of recorded publications over the years – from comprehensive model development to mapping, quantification, and monitoring of patterns and dynamics (Thomas and Wiggs, 2008; Livingstone et al., 2007; Stout et al., 2009; Zheng et al., 2022). Surface topography

characterization using high-resolution data and remote sensing such as Terrestial Laser Scanning (TLS), Light Detection and Ranging (LiDAR), and Unmanned Aerial Survey (UAS) has led to the development of quantitative methods used for monitoring purposes (Kasprak et al., 2019). Among these, UAS platforms have gained more traction due to affordability and user-friendly interface compared to other surveying counterparts. The advances in the use of UAS and Structure from Motion (SfM) photogrammetry have made geomorphic change monitoring and sediment budget estimations to become

manageable approaches in research and practice (Wheaton et al., 2009a). SfM photogrammetry utilizes a structured acquisition of images to reconstruct 3D scene geometry and camera motion based on a new generation of automated image-matching algorithms (Mancini et al., 2013). These images can be used to create point Digital Elevation Models (DEMs) to produce DEM of Difference (DoD) maps to estimate the net change in storage for morphological sediment budgets (Church and Ashmore, 1998; Wheaton et al., 2009a).

Monitoring seasonal coastal changes using high-accuracy photogrammetric images from UAVs has been widely used along the Adriatic Coast as reflected in the works of Scarelli et al. (2017), Fabbri et al. (2021), Sekovski et al. (2020), Fernandez-Montblanc et al. (2020), and Taramelli et al. (2015). Several studies have already proven the reliability of surface and elevation models from UAS-SfM on sandy coastal environments, with caution to assess the accuracy of these models before performing further analysis. The use of UAS and SfM on dune morphodynamics in Italy has not been thoroughly utilized

yet. The availability and continuous advancement of this method is an advantage that is particularly valuable for highly dynamic and vulnerable coasts, as in the case of the Emilia-Romagna seashore selected for this study.

### 1.1 Study site information

The dunes along the Ravenna coast (Northern Adriatic Sea, Italy; Figure 2) have been subjected to degradation due to combined natural, anthropogenic, and climate-induced pressures. Ravenna is an historical city, known for its beach tourism

and for having one of the largest seaports in Italy. It is part of the 130-km coastline of the Emilia-Romagna made up of flat alluvial sandy system, with gently sloping seabed of about 6 m in depth and shallow subtidal sediments from well-sorted fine to medium sand (Airoldi et al., 2016; Harley et al., 2016). The local hydrodynamic conditions include exposure to moderate wave action and a microtidal regime that ranges between 30 and 80 cm between neap and spring tides (Biolchi et al., 2022). Two wind patterns prevail in the region – the Bora wind from the northeast that brings shorter, but energetic waves and the

long-wave inducing Sirocco from the southeast. The wave climate and current circulation in the northern Adriatic are known to be strongly influenced by the Bora wind given the coast orientation.





According to the 2016 to 2020 meteo-marine data from the Hydro-Meteo-Climate Report of the Regional Agency for Prevention, Environment and Energy of the Emilia-Romagna Region (Arpae, 2020), majority of the stronger waves (0.2 m to 4 m) are from NE and ENE (Figure 1). Waves blown from the eastern side are more frequent from 2016 to 2020 but are relatively weaker (0.2 m to 2.5 m). The number of storm surges per year ranges from 17 to 24, with an average duration of 12.8 to 27.9 hours. Wind and wave data are recorded from the wave buoy every 30 minutes and are then archived to the Arpae service database that can be web accessed through Dext3r (https://simc.arpae.it/dext3r/). Historical records of the storm surge characteristics from the 2007-2020 observations are summarized in Table 1.

**Table 1: Storm surge characteristics from 2016 to 2020 extracted from Arpae database.**

| Year | # of storm surge | Total duration (h) | Ave. duration (h) | Normalized energy (m²h) | Ave. SWH (m) | Max SWH (m) | Max SL during storm surge (m) |
|------|------|------|------|------|------|------|------|
| 2016 | 23 | 343 | 14.9 | 55.1 | 1.80 | 3.11 | 0.93 |
| 2017 | 17 | 325 | 19.1 | 95.9 | 1.89 | 3.68 | 0.87 |
| 2018 | 15 | 419 | 27.9 | 111.4 | 1.88 | 3.10 | 1.06 |
| 2019 | 24 | 307.5 | 12.8 | 41.8 | 1.67 | 2.10 | 1.16 |
| 2020 | 18 | 340.5 | 18.9 | 76.3 | 1.85 | 3.11 | 1.03 |

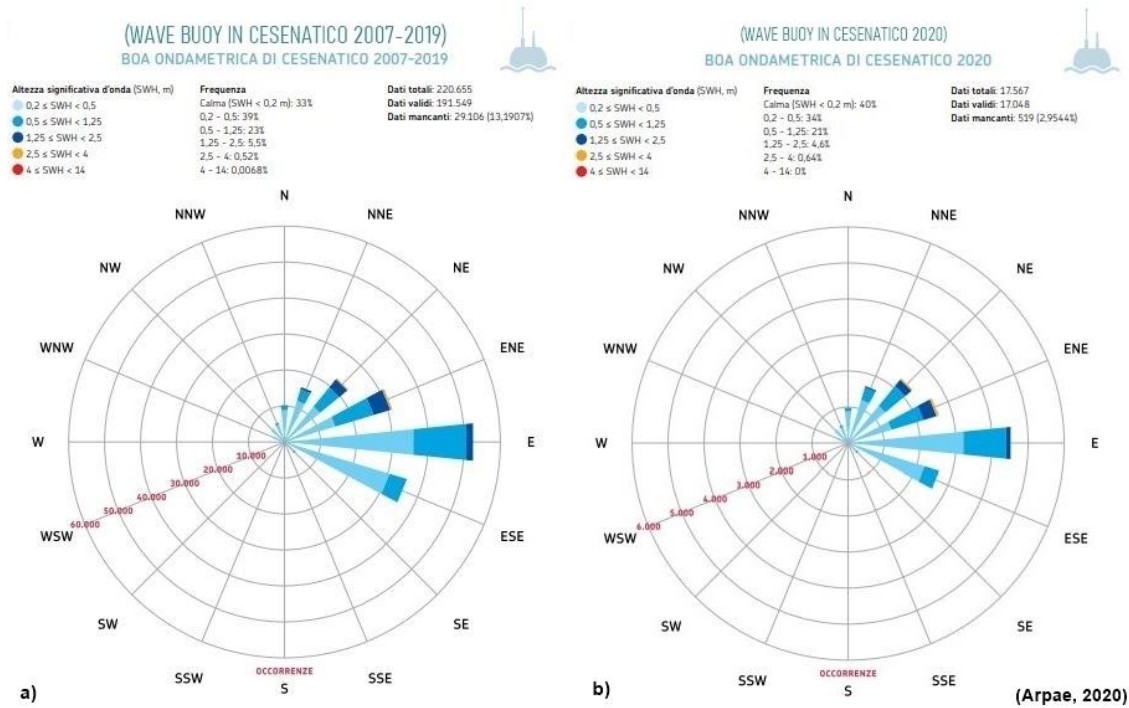

**Figure 1: Significant wave height (SWH in m) and frequency (%) for 2007-2019 (a) and 2020 (b) extracted from Arpae 2020 report.**



Significant land subsidence due to tectonic processes and sediment consolidation had been widespread and were intensified due to human activity since the second half of the 20<sup>th</sup> century (Airoldi et al. 2016). Erosive processes have also affected 105 km out of the 130 km coastline of the region during this period - along with the increase in vulnerability to storm surge, rapid coastal urbanization, implementation of rigid coastal defences, and massive dune destruction, igniting the need to implement strategic interventions to mitigate these problems (Arpae, 2020). Hard coastal defences (submerged and emerged

breakwaters, groynes, and revetments) were constructed in the early years (Armaroli et al., 2019; Perini et al., 2017). However, these infrastructures had negative environmental impacts including increased sedimentation of silts and clay, loss of native habitats, eutrophication, and poor water quality (Airoldi et al., 2016, Preti et al., 2010; Sekovsi et al., 2020). Progressive transition from hard coastal defences in the 1970s to more integrated approaches with soft techniques and Nature Based Solutions (NBS) in recent years were implemented in the region. NBS and soft-engineering techniques such as beach

nourishment and dune fence installation have been eventually initiated as alternative solutions in early 2000. Collaborations between the regional environmental agency Arpae, research groups, and other regional services that deals with coastal management led to the collection of important databases that aided the implementation of several policies to address the impending issues along the Emilia-Romagna coasts.

In 2016, the RIGED-RA Project – 'Restoration and management of coastal dunes along the Ravenna coast' was able to

install a set of 465 m wooden fences along a portion of the Bevano River Dune Ridge in Lido di Classe (Figure 2) as an intervention strategy to reduce the vulnerability of the coast and the associated residual dunes in the area (Giambastiani et al., 2016). The Bevano dune-beach system is a protected natural area with high biodiversity, with laterally continuous and sub-vertical foredunes. According to Giambastiani et al. (2016), the area was selected as the pilot site given that it has the potential for dune accumulation but has limited beach width and unstable sub-vertical foredune geometry. The first fence

was placed at the dune foot followed by the second fence 2 m seawards. The two fences were connected perpendicularly by 8 m fence portions. No planting activities were implemented due to high presence of native sand-binding vegetation species such as the *Psammophilous*.



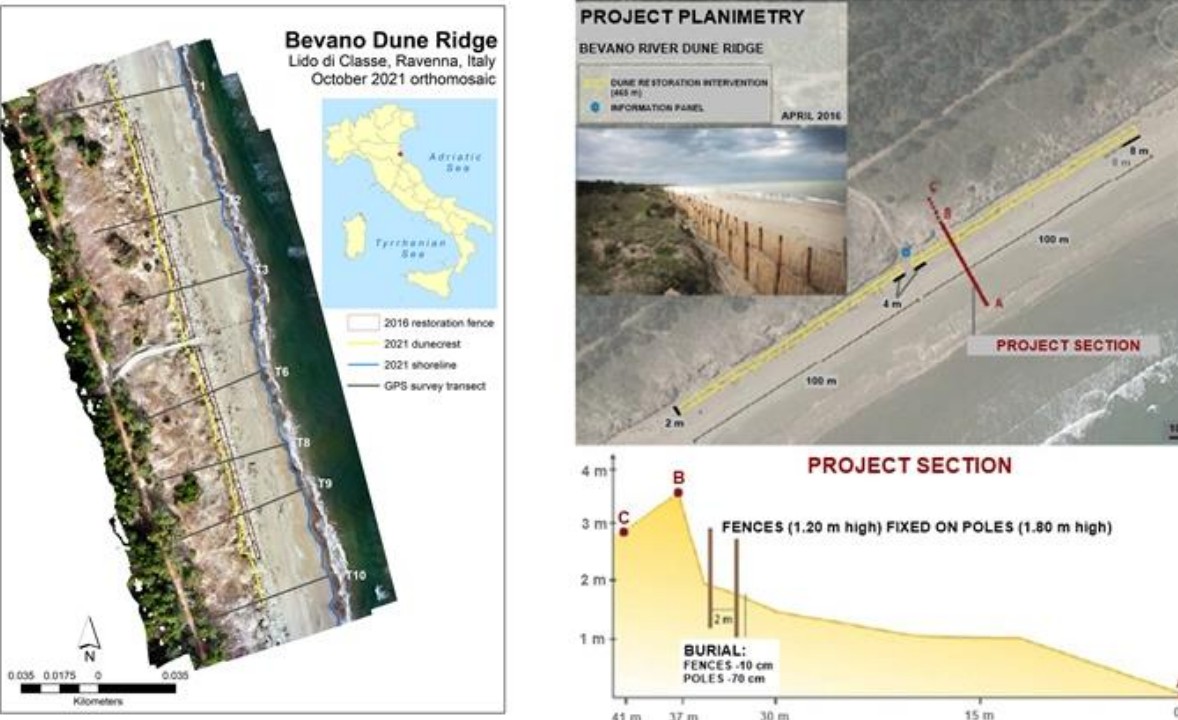

**Figure 2: Location of the study area in Ravenna (Italy), and the dune fence project planimetry (modified from Giambastiani et al., 2016).**

The availability of repeated UAV topographic surveys after the fence installation and the availability of open-source tools can address the gaps in quantifying the restoration efficacy. The study aims to provide informed decision from qualitative data analysis with the proposed workflow for UAV data processing and elevation data analysis suited for sediment volume calculation. Error analysis was performed to validate the produced change detection results. Vegetation change using orthomosaic images derived from UAV were also explored to determine other contributing factors to the overall morphology of the dune ridge.

## 2. Materials and Methods

### 2.1 UAV survey and elaboration

The methodological framework (Figure 3) includes established workflows for data acquisition, geomorphology modelling, vegetation change, and geomorphic change detection. Annual monitoring campaigns were carried out after the fence installation (2016); the October 2016 and 2021 UAV and GPS surveys were selected to assess the dune evolution. GPS data points were collected using a Leica DGPS (Viva GNSS GS15 GPS) that worked with a Real Time Kinematic (RTK) system to ensure sub metric accuracy. Collected datapoints include several profiles across the beach from the coastline to the back dune. Aerial photographs for the 3D reconstruction were captured using a DJI Phantom drone, with flight plans defined in





the UAV software and image overlap of 70%. Targets were established on site and were geolocated using GPS that were used to georeference the images during data processing and DEM development.

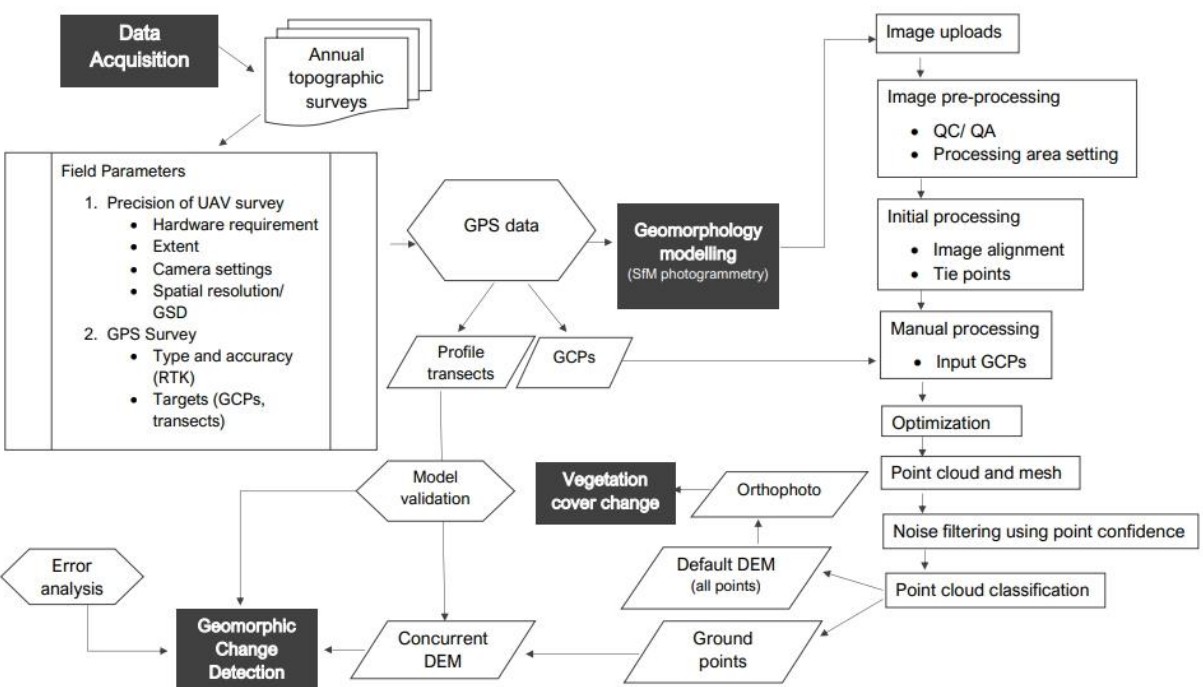

**Figure 3: Methodological Framework of the study.**

The geomorphology modelling was performed by SfM processing in Agisoft Metashape Professional. Multiple overlapping

photos were loaded, and initial image filtering was performed, wherein only images with quality value of >.50 units based on the sharpness level are retained. Photos from take-off and landing were also removed before the image alignment process. Ground Control Points (GCPs) were used to georeference the UAV point cloud; each GCP was assigned as either a control or check point - the former is used to reference the model, while the latter is used to validate the camera alignment accuracy and optimization procedure results (Agisoft LLC, 2021). The 70-30 rule was implemented in selecting the control and

checkpoints – 70% of the total GCPs were used as control points while 30% were checkpoints. Camera alignment was then performed to improve the accuracy, followed by the depth maps and dense point clouds generation. Point cloud noise removal was done by implementing filtering by confidence, where points with confidence value from 0-5% were removed. The remaining dense point clouds were classified to automatically divide all points into two classes – ground points and others using the automatic classification of ground points. Based on Agisoft LLC (2021), dense cloud is initially filtered for

noise and is divided into a set cell size. The lowest point is detected in each cell then first approximation of the terrain is done from the triangulation of these points. A new point is added to the ground class if it satisfies the set conditions in terms of a certain maximum angle and distance, and this is repeated until all points are checked. The default values were used to



set the parameters. All classified points were used in the automatic DEM creation in the software that is used to create the orthomosaic, while the classified ground points were used to create the DEM for the change detection part. Same values and spatial resolution of 0.1 m x 0.1 m in all parameters were applied to both 2016 and 2021 data to ensure coherence and comparability.

The geomorphic change assessment was implemented by GCD 7 AddIn for ArcGIS 10.x toolset in which the difference between the DEMs is calculated to provide a spatially continuous estimate of the topographic changes within the study area (Kasprak et al., 2019). Geomorphic Change Detection (GCD) is a tool developed by Riverscapes Consortium using the methodology of Wheaton et al. (2009a), Wheaton (2008) and Wheaton et al. (2009b). It can compute for the extent, magnitude and landscape form changes that occur within an inter-survey period to understand the overall sediment budget of the area of interest or the spatial distribution changes of sediments through time (Grams et al., 2015; Collins et al., 2016; Sankey et al., 2016). An Area of Interest (AOI) that consisted of the foredune to beach area was also used as the processing extent. Error rasters were created using the reported control point error values in the SfM processing, which were 0.046 m and 0.032 m, respectively. The values were used to calculate the propagated error applied to each cell (Eq. 1) and the T-statistics (Eq.2) adapting the following equations from Lane et al. (2003):

$$\sigma_c = \sqrt{\sigma_1^2 + \sigma_2^2} \tag{1}$$

where $\sigma_c$ is the root sum of square of uncertainty for each change interval [m]; $\sigma_1^2$ and $\sigma_2^2$ are the squares of uncertainty for the older and newer time steps [m];

$$t = \frac{z_{t2} - z_{t1}}{\sigma_c} \tag{2}$$

where $t$ is the t-statistics, $z_{t2}$ and $z_{t1}$ are the elevation of the raster cell for the newer ($t_2$) an older ($t_1$) time step [m], respectively. The T-statistics can be used as a thresholding level of significant change, where values of t > 1.96 mean confidence interval of 95% for a two-tailed t-test. The values that fell below the confidence threshold were removed from the output change raster that improved the likelihood that a significant change was captured (Hilgendorf et al., 2021). The percent sediment imbalance metric *SI* was also calculated to characterize the sediment dynamics using Equation 3 (Wheaton et al., 2013; Kasprak et al., 2015; Kasprak et al., 2019):

$$SI = \frac{(V_{DEP} - V_{EROS})}{2*(V_{DEP} + V_{EROS})} * 100 \tag{3}$$

where $V_{DEP}$ is the volume of deposition or the positive topographic change and $V_{EROS}$ is the volume of erosion or the negative topographic change [m³].

The validation of the resulting Digital Elevation Models (DEMs) was performed by comparing the GPS profiles carried out in the field with the elevation values extracted from the models using the QGIS Profile Tool plugin version 4.2.0. Validation of the model and field elevation values were visualized and regressed in Python 3.9; statistical measures include $R^2$, Root-Mean-Square-Error (RMSE), Mean Absolute Error (MAE) and the Mean Bias Error (MBE).



The vegetation change analysis was performed based on the methodology by Silvestri et al. (2022), with some modifications
applied. The process includes shoreline delineation using ISP cluster unsupervised classification in ArcGIS, transect creation
based on selected GPS profiles, gridding, and centroid creation at 1m x 1m resolution. Each centroid was manually assessed
and classified according to its dominant cover feature that could either be vegetation, logs or debris, and bare sand. Point
density of the cover types in each transect were used to come up with a percentage calculation to represent the change from
the 2016 and 2021 orthomosaic images.

## 3. Results

### 3.1. DEM development and validation

The DEMs, at 0.1 m x 0.1 m resolution, resulting from the UAV surveys in 2016 and 2021 are shown in Figure 4. There is
an elevation range of -0.37 to 6.30 m, with the minimum and maximum values observed along the beach area and the back
dunes. Validation on the sample transects (Figure 4b) was performed using regression analysis shown in Figure 5. Only the
2021 survey was validated since there are no ground-truth GPS profiles available for the 2016 dataset. The $R^2$ values range
from 0.97 to 1, while the RMSE values range from 0.07 m to 0.15 m. The MAE and error bias values range from 0.06 m to
0.10 m and -0.01 m to 0.05 m, respectively.

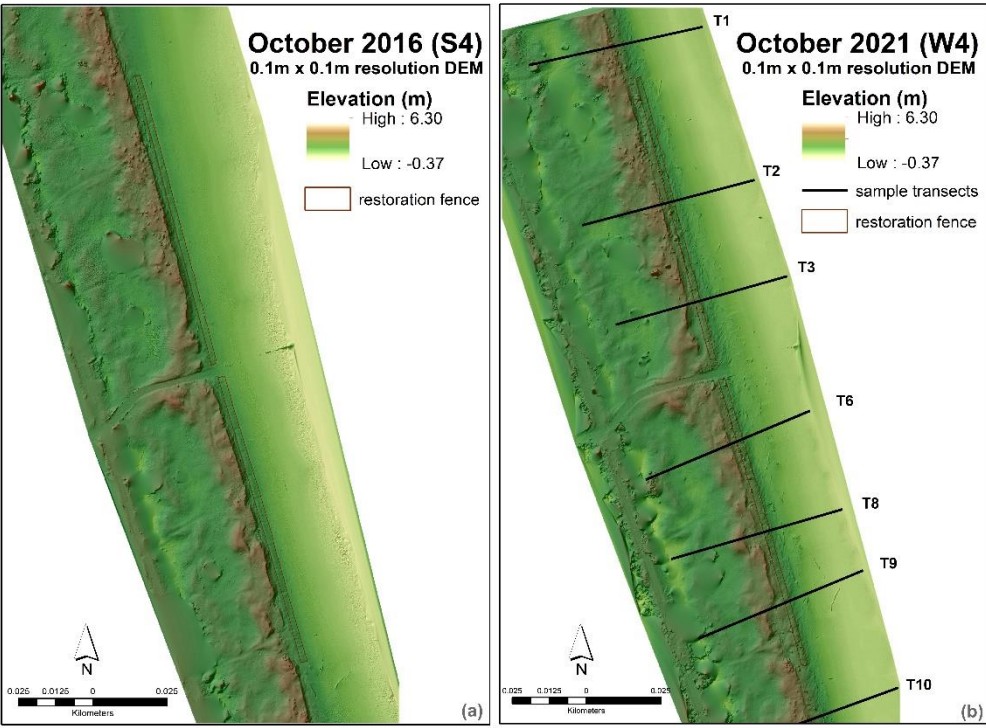

**Figure 4: DEM of 2016 (a) and 2021 survey with GPS profiles (b).**






**Figure 5: GPS vs DEM profile comparison for 2021.**

## 3.2 Geomorphic and vegetation cover changes

Figure 6 and Figure 7 show the change detection result of 2021 vs 2016 DEMs; considering the total area of 9154 m$^2$, we
found that 6020 m$^2$ has detectable changes after applying the 95% C.I. threshold, representing 66% of the area of interest.
Specifically, 2221 m$^2$ had surface lowering while 3799 m$^2$ had surface raising. For volumetric change, there is a total of 584
m$^3$ surface lowering and 1109 m$^3$ surface raising. Therefore, a net rise of 1692 m$^3$ has been detected, with a net volume
difference of + 525 m$^3$. An average depth of 0.26 m and 0.29 m of surface lowering and raising was calculated in terms of



vertical averages. The average total thickness of difference is 0.18 m, with a net thickness difference of 0.06 m. In summary,
34% elevation lowering, and 66% surface raising were calculated in terms of percentage by volume, with a percent
imbalance or departure from the equilibrium of 16%. A tabular summary of the change detection is included in the
Appendix. DEM profile comparison was also performed to show the sand deposition and erosion along the surveyed
transects.

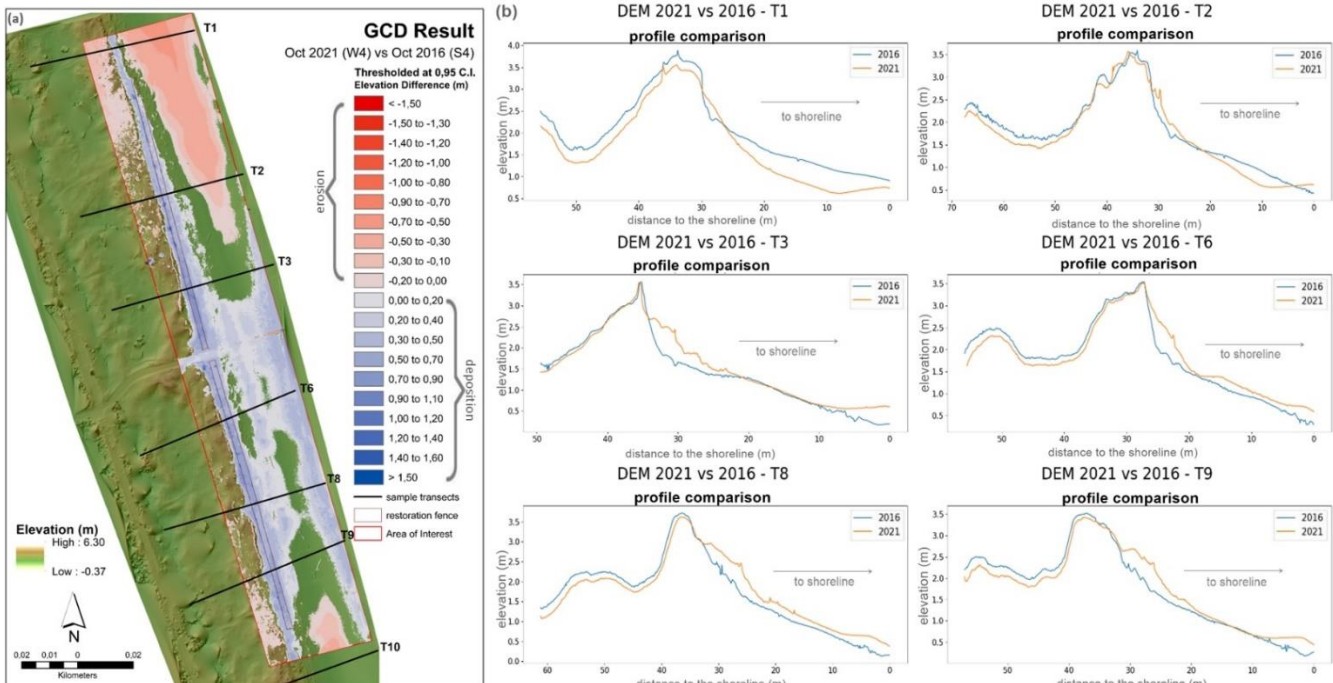

**Figure 6: Change Detection result of 2021 vs 2016 DEMs (a); and DEM transect profile comparison (b).**

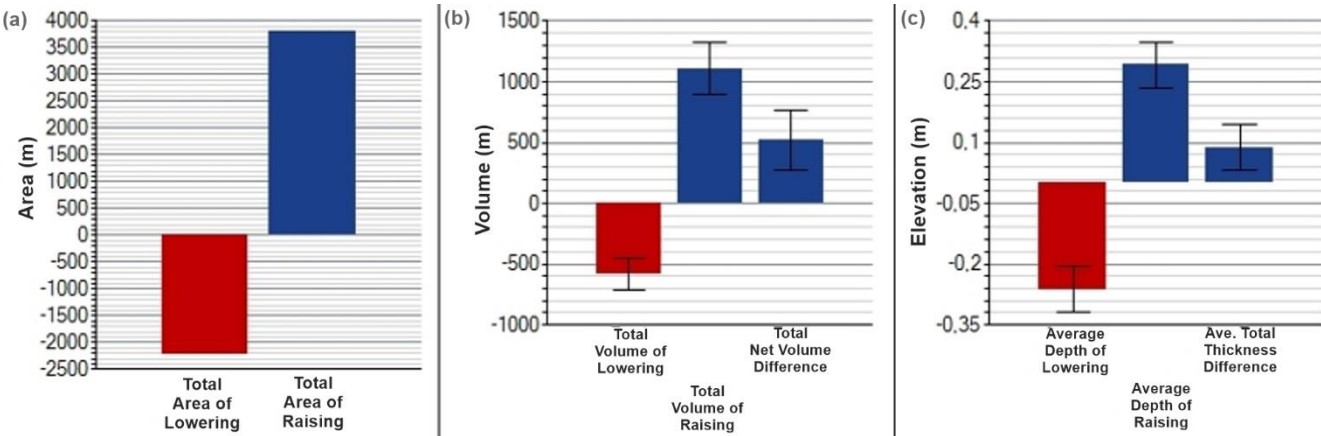

**Figure 7: Summary of the areal (a), volumetric (b), and elevation changes (c).**





2016 and 2021 orthomosaic images were used in the vegetation cover change assessment (Figure 8). There was an overall increase in the cover extent of vegetation and areas with logs or debris and consequently, an obvious decrease in bare sand extent. The highest positive percent change for vegetation increase were in transects 2, 3, 8, and 9. Increase in logs and debris was more evident in transects 2, 3, 6, 8, and 9 (Figure 9).

**Figure 8: Vegetation cover change maps between 2016 (a) and 2021 (b).**





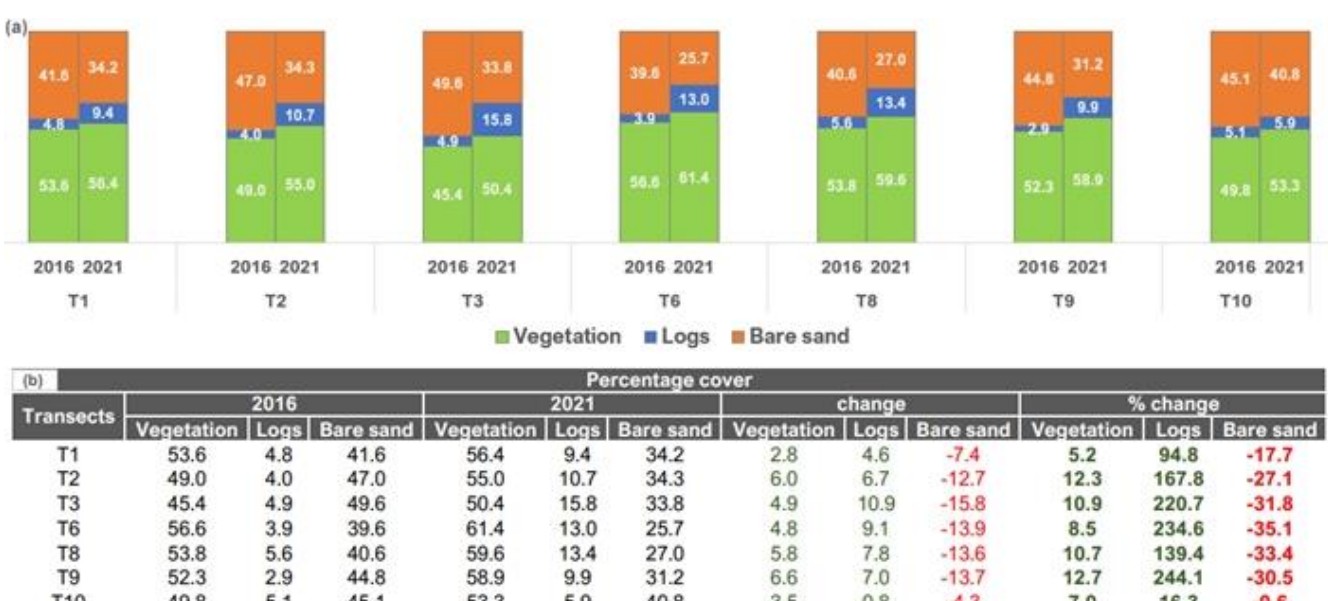

Figure 9: Percentage cover comparison (a) and percent change (b).

## 4. Discussion

### 4.1 Change detection - geomorphic and vegetation cover

The GCD result highlights a significant deposition – in terms of area, average depth, and volume, along the dune foot and portion of the beach. The overall foredune stoss slope has reduced given the sediment accumulation and the formation of embryo dunes within the fence. Progradation of around 3 to 5 m from the foredune has been evident and some profiles exhibit embryo development (Figure 6b, profiles 3, 6, 8, and 9). Profiles with significant sand volume increase were the ones located at the middle and the southern part. Embryo foredunes are formed due to sand deposition within relatively clumps of vegetation, individual plants, or driftwood/ log debris (Hesp, 2002; van Puijenbroek et al., 2016). Increase in vegetation and driftwood has been observed within and near the fence (Figure 8). Most of the vegetation change appeared in between the fences as pioneer species colonized the pillow sand deposits. No vegetation change on the more stabilized back dune was observed. The increase in vegetation colonization has contributed to the stabilization of sand accumulation within the dunes over 5 years, which was also evident in the study of van Puijenbroek et al. (2016). Tolerant vegetation facilitates sand deposition and aid in increasing stabilization and growth of dune systems (Laporte-Fauret et al., 2021; Wooton et al., 2016). The result is also in agreement with the published work of Dong et al. (2008) and Hesp (2002), where it was mentioned that the establishment of vegetation on bare sand or beach forms a roughness element that may allow localized sand deposition and reduced erosion.

An erosional pattern is apparent in the northern beach portion towards the northern head of the structure (Figure 6a), which may be accounted for by the aerodynamic and morphodynamic conditions around the dune fence, the efficiency of the fence



and its configuration to trap sediments. The effects of sand trapping fences are primarily determined by its geometry,
orientation relative to the main wind direction, aerodynamic roughness of the wind profile, undisturbed flow, and shelter
distance (Eichmanns et al. 2021). Its influence is dependent on the given sediment boundary conditions and the wind field. In
this case, the northern beach portion and the fence peripheries have relatively lesser debris and vegetation change which
could have caused the erosion along the beach.

No significant increase in the foredune heights have been evident in all the profiles (Figure 6b). Similar findings have been
observed in the study of Itzkin et al. (2020), where new embryo dunes are created seaward of the original dune following the
emplacement of sand fences, but this has impeded the natural foredune from receiving additional sediment. Hence, dune
fencing may not always be the best singular management action as it can also prevent deposition to the natural dune behind
the fence that could limit vertical growth.

Human disturbances by mass tourism and coastal urbanization put detrimental impact on pioneer vegetation species and can
prejudice the sustainability of foredune restoration especially in the Mediterranean (Della Bella et al., 2021). The fence has
halted human trampling on dune vegetation that had limited the formation of erosional features such as blowouts. The
enforcement of limited human disturbance and the fact that it is part of a protected area had supported the restoration
efficacy and the spontaneous recovery of the dunes. Compared to beaches that are mechanically cleaned for recreational
purposes, driftwoods that were not removed in this area acted as a form of passive restoration as well. These soft-engineering
and nature-based solutions were able to induce sand accretion and vegetation colonization that can be considered as
ecologically sustainable, technologically feasible, and economically viable. Given the restoration results, dune fencing and
limiting debris cleaning can also be implemented along the other coastal zones of the Emilia-Romagna and other low-lying
sandy coasts as it can aid in the sediment exchange system over time since both sediment contribution from the nearshore to
the dunes and accretion rates in the foredune are vital in a beach-dune system.

**4.2 Error analysis**

The accuracy of the change detection model heavily depends on the quality of the input DEMs. Precision issues with the
SfM-derived DEM and GPS have been encountered due to target data quality and systemic issues. The model fitting
statistics of the 2021 dataset show a good fit within 0.96 to 0.99. However, a slight shift of values in the back dune is evident
that can be accounted to human error during the GPS survey as the pole can be dragged a few centimeters from the ground.
Possible variance between in beach surfaces, systematic collection inconsistencies related to survey set-up and susceptibility
to external factors, such as possible digging of the pole and wind speed influence may be encountered on beach
environments surveys (Talavera et al., 2018; Casella et al., 2020; Hilgendorf et al., 2021). Misclassification of vegetation to
ground points may have also affected the accuracy of the surface reconstruction of the DEMs. Drone-based topographic
reconstructions of beach environments tend to exhibit higher inaccuracies compared to other environments such as outcrops
due to low texture and contrast of sand surface, making photogrammetric methods, such as features matching, difficult



(Eltner et al., 2015; Casella et al., 2020). Another probable source of error is the lack of validation information for the 2016 data that might have affected the accuracy of the change detection model.

Notwithstanding, the results show that assessing the spatio-temporal evolution of the erosional and depositional processes in the Bevano dune ridge is possible using multi-temporal drone data. Elevation model accuracy in the order of ~5 to 8 cm has been achieved. The results of the study may be further improved by ensuring consistency in camera and build parameters for the elevation and change detection models. Classifying the area of interest into geomorphic units can also be done to enhance the geomorphic change detection result.

## 5. Conclusion

A dune restoration project in the Northern Adriatic coast (Ravenna, Italy) was assessed using UAV monitoring surveys. SfM photogrammetry, elevation differencing, and statistical analysis were utilized to quantify dune development in terms of sand volume and vegetation cover change over time.

Despite the natural factors affecting the overall deposition dynamics in the area, results show that dune fencing proved to be an effective intervention to prevent dune erosion since significant geomorphological changes and vegetation colonization occurred in the 2016-2021 interval time. Main sand accumulation was observed along the dune foot where the wood fences were established. The following changes have also been observed: progradation of the front dune; development of embryo dunes; decrease in slope stoss; decrease of blowout features due to increase in vegetation colonization; and increase in vegetation and debris cover within and near the wood fences.

GCD can be an effective monitoring tool for coastal dunes for as long as the sources of uncertainties are considered. The results of the study can supplement in showcasing the importance of implementing dune fencing and limiting debris cleaning as nature-based solutions to prevent dune degradation along the coastal zones of the Emilia-Romagna. The proposed systematic workflow developed within this research can be transferred to other similar coastal zones and implemented into guidelines for Integrated Coastal Zone Management (ICZM).



**Appendix A:**

| Attribute | Raw | Thresholded DoD Estimate: | | | Description |
|---|---|---|---|---|---|
| **AREAL:** | | | | | **AREAL METRICS** |
| Total Area of Surface Lowering (m²) | 4,168 | 2,221 | | | The amount of area experiencing a lowering of surface elevations |
| Total Area of Surface Raising (m²) | 4,986 | 3,799 | | | The amount of area experiencing an increase of surface elevations |
| Total Area of Detectable Change (m²) | NA | 6,020 | | | The sum of areas experiencing detectable surface elevation changes |
| Total Area of Interest (m²) | 9,154 | NA | | | The total amount of area under analysis (including detectable and undetectable) |
| Percent of Area of Interest with Detectable Change | NA | 66% | | | The percent of the total area of interest with detectable changes (i.e. either exceeding the minimum level of detection or with a proability greater then the confidence interval chosen by user) |
| **VOLUMETRIC:** | | | ± Error Volume | % Error | **VOLUMETRIC METRICS** |
| Total Volume of Surface Lowering (m³) | 696 | 584 ± | 124 | 21.32% | On a cell-by-cell basis, the DoD surface lowering depth (e.g. erosion, cut or deflation) multiplied by cell area and summed |
| Total Volume of Surface Raising (m³) | 1,177 | 1,109 ± | 213 | 19.20% | On a cell-by-cell basis, the DoD surface raising (e.g. deposition, fill or inflation) depth multiplied by cell area and summed |
| Total Volume of Difference (m³) | 1,873 | 1,692 ± | 337 | 19.93% | The sum of lowering and raising volumes (a measure of total turnover) |
| Total Net Volume Difference (m³) | 481 | 525 ± | 247 | 46.99% | The net difference of erosion and deposition volumes (i.e. deposition minus erosion) |
| **VERTICAL AVERAGES:** | | | ± Error Thickness | % Error | **VOLUMETRIC METRICS NORMALIZED BY AREA** |
| Average Depth of Surface Lowering (m) | 0.17 | 0.26 ± | 0.06 | 21.32% | The average depth of lowering (surface lowering volume divided by surface lowering area) |
| Average Depth of Surface Raising (m) | 0.24 | 0.29 ± | 0.06 | 19.20% | The average depth of raising (surace raising volume divided by surface raising area) |
| Average Total Thickness of Difference (m) for Area of Interest | 0.20 | 0.18 ± | 0.04 | 19.93% | The total volume of difference divided by the area of interest (a measure of total turnover thickness in the analysis area) |
| Average Net Thickness Difference (m) for Area of Interest | 0.05 | 0.06 ± | 0.03 | 46.99% | The total net volume of difference divided by the area of interest (a measure of resulting net change within the analysis area) |
| Average Total Thickness of Difference (m) for Area With Detectable Change | NA | 0.28 ± | 0.06 | 19.93% | The total volume of difference divided by the total area of detectable change (a measure of total turnover thickness where there was detectable change) |
| Average Net Thickness Difference (m) for Area with Detectable Change | NA | 0.09 ± | 0.04 | 46.99% | The total net volume of difference dividied by the total area of detectable change (a measure of resulting net change where the was detectable change) |
| **PERCENTAGES (BY VOLUME)** | | | | | **NORMALIZED PERCENTAGES** |
| Percent Elevation Lowering | 37% | 34% | | | Percent of Total Volume of Difference that is surface lowering |
| Percent Surface Raising | 63% | 66% | | | Percent of Total Volume of Difference that is surface raising |
| Percent Imbalance (departure from equilibrium) | 13% | 16% | | | The percent departure from a 50%-50% equilibirum lowering/raising (i.e. erosion/deposition) balance (a normalized indication of the magnitude of the net difference) |
| Net to Total Volume Ratio | 26% | 31% | | | The ratio of net volumetric change divided by total volume of change (a measure of how much the net trend explains of the total turnover) |

**Table A1: Tabular summary of the change detection analysis.**

**Author contribution**

Conception and study design: RAF, BMSG; Data collection: RAF, BMSG, LC; Data analysis and interpretation: RAF, BMSG, LC, SS; Article drafting: RAF; Critical revision of the article: RAF, BMSG, SS; Final approval of the version to be published: BMSG, SS, LC.

**Competing interests**

Some authors are members of the editorial board of the current special issue "Monitoring coastal wetlands and the seashore with a multi-sensor approach". The peer-review process was guided by an independent editor, and the authors have also no other competing interests to declare.

**Acknowledgements**

Regine Anne Faelga and the collaboration among the authors were supported by the European Commission under the Erasmus Mundus Joint Master Degree Programme in Water and Coastal Management [grant number 586596-EPP-1-2017-1-IT-EPPKA1-JMDMOB].



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
