# Peer review of "Dune belt restoration effectiveness assessed by UAV topographic surveys (Northern Adriatic coast, Italy)"

_EGUsphere, 2023_

## Author Comment (AC4)

| Reviewer 1 | Authors |
|---|---|
| **GENERAL COMMENT:** In this manuscript, drones and GPS were used in two surveys (2016 and 2021) to assess the efficiency of using fences to restore the Bevano dune ridge (Ravenna, Italy). The DEMs and orthomosaics obtained were used to compute a DEM of difference and to classify the area in 3 classes, respectively. The previous allowed the analysis of dune geomorphic and vegetation changes over time. At its present form the manuscript is not ready to be published, and I suggest the authors to perform major changes. The intention of this review is to help them to improve the manuscript, and I hope they find the comments constructive. The different sections are not well connected, as the authors do not introduce properly some aspects (e.g. blowouts, dune slopes) neither in the introduction nor the methodology, but then discuss about those all of a sudden. In addition, important aspects of the methodology (e.g. drone flight elevation, number of control points and configuration used, or image classification accuracy assessment) were not performed, or at least not explained in the text. In addition, the result and discussion sections are a bit vague (see specific comments below). On the other hand, figures should be carefully revised and improved, mainly increasing the size of the font used as they are hard to read, adding the correct units, and adequate size bars and scales. I also recommend the authors to re-phrase some paragraphs and make them easier to read and follow, avoid repetitions, and to review the English, as there are typos throughout the text. | Thank you for providing a very constructive review. We recognize the need to revise portions of the manuscript to improve its overall flow and readability.

We have already implemented revisions in the figures and are working on revising the rest of the manuscript to include aspects that have been missed in the uploaded version. Drone flight survey information (flying height, number of control points, and configuration) will be included in the Materials and Methods. Updated information regarding the image classification used for the cover analysis will also be provided to give context on how it was implemented. All the specific comments per section are considered and will be reflected in the updated version.

Rest assured that all figures will be carefully revised, and the paragraphs will be improved. |

| Specific comments | |
|---|---|
| **Abstract:**

Lines 8-10: remove that phrase since it would be better placed at the introduction, it does not say much at the abstract. | The phrase will be removed from the abstract and will be placed at the introduction. |
| Line 11: what's the temporal scale of your study? It is not mentioned but it is important to indicate it, as you later on talk about dune progradation of 3-5 metres. Also, how many drone surveys were performed? | The temporal scale is 5 years using two surveys (October 2016 and October 2021). This information has been stated in the introduction but will be included in the abstract. |
| Line 14: Do the fences really prevent dune erosion? Or instead they promote dune recovery and growth? | Yes, the fences have been proven to prevent dune erosion. The study by Perini et al. (2015) DOI: 10.5150/cmcm.2015.044 will be included in the introduction to support this statement.

This line will also be revised to include that dune fences prove to be effective in promoting dune recovery and growth since significant sand deposition of around 1109 $m^3$ is observed mostly along the dune foot and front based on the change detection analysis. Progradation measurement (~3-5 meters) is mentioned in the next sentence, which were quantified based on the DEM transect profile comparison. |
| Lines 14-16: I suggest the authors to include numbers when mentioning the results: how much sand accumulation was measured and how the vegetation cover changed, instead of mentioning 'increase in vegetation' or 'significant sand accumulation'. | Thank you for this feedback. We will include the quantities in the final version. |
| Line 19: I would remove the line about the GCD toolset. | The line about GCD will be removed as advised. |

| | |
|---|---|
| **Introduction:**

Lines 29-30: Check the order of some references in the text, I'm not sure they are placed correctly. | Reordering of the references mentioned will be applied. |
| Line 34: Not sure these are the best references. | The reference list will be reviewed, and necessary changes will be applied in the revised manuscript. |
| Line 36: For consistency, it is better to use only one term. Before the authors were using 'UAV' and now 'UAS' | The term UAV will be used throughout the text. UAS will be replaced in the final version. |
| Lines 45-49: it is a bit repetitive | These lines will be revised and the additional literature that has been provided in Line 49 comment will be included. |
| Line 49: There are already some studies in the Emilia-Romagna region, please check the paper of Duo et al (2021): https://doi.org/10.3390/rs13091823 | Thank you very much for providing this useful resource. Line 49: "The use of UAS and SfM on dune morphodynamics in Italy has not been thoroughly utilized yet." will be revised to include the study of Duo et al. (2021) in the discussion. |
| Figure 1: The legends are hard to read. | The legends in Figure 1 have already been revised. |
| Lines 92-95: there is no mention about the characteristics of the blowouts in the area, which are mentioned in the abstract but not here. | Information about the blowouts characteristics in the area will be added to the manuscript. |
| Figure 2: Legends and numbers are really small. Also, what are the red dots (A, B, and C) indicating? | Figure 2 has been revised to improve the readability of the legends. Points A, B, and C represent areas in the back dune, foredune, and beach along a section of the project. This information will be included in the figure caption. |

| | |
|---|---|
| Line 101: how frequent were the drone surveys? | Annual drone surveys for monitoring were done after the installation in 2016. In this paper, we do not analyze the yearly surveys but only the years 2016 and 2021.

Line 101 revised to: The availability of annual UAV topographic surveys after the fence installation in 2016 and the availability of open-source tools can address the gaps in quantifying the restoration efficacy. |
| Line 102: qualitative data analysis? | This should have been "quantitative" instead of "qualitative" data analysis. Revision will be reflected in the final manuscript. |
| **Materials and methods:**

Lines 114-116: which UAV software? what was the flight elevation? 70% of front and side overlap? How were the targets distributed in the area and how many did you use? In figure 3 targets are referred to as GCPs, use the same term for consistency. | The software used for the drone flight planning and execution was Pix4DCapture. Flying heights that were used in the survey are both 20m. Upon verifying the project files, the side and front overlap is at 80% instead of 70%. The term GCPs will be uniformly applied. All this information will be included in the revised manuscript. |
| Lines 120-121: not important, I would remove these lines | These lines would be removed in the final manuscript. |
| Line 125: how many points did you use? Check points are also used to assess the horizontal and vertical accuracy of the 3D models, how did you assess those? These lines are a bit hard to understand, please re-phrase. Also, how did you remove the vegetation from the DEM?

Lines 125-134: I think there is too much info that is not needed here | For the 2016 data, a total of 13 GCPs were utilized. 9 were considered control points and 4 GCPs were checkpoints. The 2021 data has a total of 15 GCPs - 10 were controls, 5 were checkpoints.

The horizontal and vertical accuracy of the 3D models were assessed using field GPS comparison and the calculated RMSE of the selected checkpoints. These details will be included.

The vegetation from the DEM was removed using the automatic ground point classification (using the default parameters, which will be added in the text or annex) of |

| | the Agisoft Metashape software. Basically, non ground points were considered the vegetation and other structures that are present in the area. Only the points that were classified as ground were utilized to create the concurrent DEMs.

The lines mentioned in this comment will all be rephrased accordingly. |
|---|---|
| Line 137: ArcGIS 10.x? | This line has been changed to the software version used, which is ArcGIS 10.8.2 |
| Lines 125-163: I suggest to simplify and re-phrase these lines, they are a bit confusing and hard to follow as they are right now. | This suggestion will be implemented in the final manuscript. This part will be rephrased and simplified to be better understood. |
| Line 166: why 1 x 1 m of grid resolution? Also, how was the accuracy of the unsupervised classification assessed? This is important but it is not mentioned. | The method applied to detect the presence/absence of vegetation is based on a statistical approach similar to Silvestri et al. (2022) and is based on the visual inspection of the centroids of the grid cells: if the centroid falls on bare soil we have absence of vegetation while if it falls on a vegetated pixel of the photo, we have presence of vegetation. As the orthophotos have a resolution of 0.1 m x 0.1 m, a 1 x 1 m grid resolution allows us to sample one pixel (corresponding to the centroid) every 100 pixels included in each grid cell. This method is similar to a classic visual ecological survey performed in the field with 0.1m x 0.1m plots placed at a distance of 1m from each other along a transect, but in this case it is performed on an orthophoto instead, with the assumption that the operator has a clear overview of the area and can clearly distinguish between vegetated and non-vegetated (either with bare sand or logs/debris) pixels.

Therefore, the accuracy of the method depends on the ability of the operator as well as on the image quality.
In the new version of the paper, we will include these details about the method and will clearly state the assumptions. |

| | As for the classification, we decided not to apply any classification method because no specific field surveys were originally designed to collect ground truths on vegetation, vanishing any possibility of validating the results unless visual interpretation were used to perform such validation, with an approach similar to the one that we applied directly to the orthophotos in our statistics. |
|---|---|
| **Results:**

Line 175: Maybe explain where those values are located, in which profiles? help the reader a bit to understand what happened in your area of study, and especially what happened with the dune (do not forget it is the goal of your work) | This part will be improved to include more context for the values that were produced. However, this part is focused on the DEM development and validation process. The dune changes were discussed in the latter part (3.2 Geomorphic and vegetation cover changes). |
| Figure 5: profile 10 is not included. | Profile 10 has been included in the revised Figures 5 and 6. |
| Line 184: if no vertical accuracy of the 2016 DEM could be obtained, which error associated to this DEM did you propagate to obtain the DoD? only the error of the 2021 DEM? | The propagated error in the 2016 DEM includes the RMSE of the selected control points. |
| Line 185-193: These lines are a bit repetitive and confusing. It could help if you describe in which areas of the study site you found the erosion or the deposition of sand, using the map of Figure 6. Was erosion identified at the dune or the beach? Or both? Again, these results should be focused on describing the changes on the dune especially, as it is the 'goal' of your work. | Thank you for this feedback. The detailed discussion regarding the change detection was mentioned in Lines 210-228. Although, we agree that this part needs revision to better describe the dune changes using the map in Figure 6.

The provided suggestions will be considered. |
| Line 186: use erosion and deposition instead of lowering and raising, also in the figures | The authors will use erosion and deposition in the revised version. |

| | |
|---|---|
| Figure 6: the numbers and letters are very small. Also, use metres instead of km for the distance bar at the map (also in the rest of the figures). | The GCD map was enlarged, and the scale bar was changed to meters. Profile 10 was also included in the revised figure. |
| | All map scale bars were revised from kilometers to meters. |
| Figure 7: please, check carefully the units of each variable, in all the figures and the text. And also the scale of the Y axis, why the vertical scale is not consistent along the negative and positive values in the elevation graph? | Thank you for noticing the error. We have already changed the units for the areal, volumetric, and average depth of changes. We have also corrected the vertical scale and term of the average depth change (elevation). |
| | Aside from correcting the vertical scale, both surface erosion and deposition are represented by positive values to allow easier comparison. |
| | The terms lowering and raising were also revised to erosion and deposition. |
| I am missing some measurements here about the dune progradation that were mentioned in the abstract? | Progradation result was indicated in Line 210-211 (Section 4.1) |
| Line 201: what happens with profile 10? It is not included in the previous analysis but it is included in Figures 8 and 9, but no results associated to it are explained. It looks like vegetation cover did not increase that much compared to the debris or logs. At this point, I wonder whether the unsupervised classification method differentiates well both classes (logs or debris from vegetation)? Did you assess the classification accuracy?How did you do it? In the methods is not explained and neither is here, and it is quite important | Profile 10 was already included in Figures 5 and 6. This profile has been processed but was not included in the current manuscript due to layout restraint. |
| | As mentioned in the comment Line 166, the statistical approach used to assess the vegetation cover is based on the visual inspection of the 2016 and 2021 orthomosaic images. |
| | The authors will ensure that additional information pertinent to the cover analysis method will be included in the revised manuscript. |
| Figure 8: the 2021 dune crest and shoreline are shown over the 2016 orthomosaic, please be careful. Why are not the rectangles | The 2021 data was used as reference for the dune crest and shoreline delineation. |

| | |
|---|---|
| perpendicular to the shoreline? The transition between blue and green is not easy to see, maybe change the colours. | The rectangles are not perpendicular since the boundary condition used in the fishnet creation was the profile transects. We opted not to adjust the default fishnet output since the goal is to observe cover change between the 2 survey periods.

The maps were improved in terms of color, font size, scale bar, and legend details. |
| **Discussion:**

**Line 208**: significant compared to what? are any other similar areas experiencing less deposition for the given time span of 5 years? You could compare accretion values and progradation rates measured in other similar areas, in order to know if is significant or not. Maybe it would be good to explain what's the sediment budget in the area, is there enough sediment arriving or the sediment is scarce there? | The sediment budgets along the Emilia-Romagna coast are very complex and site specific. It is difficult to compare the sediment budget of a nearby area to the study area. We will include the discussion about the complex system of the Emilia-Romagna coastline focusing on this stretch of coast using the last published report of the regional agency for environment (Report of the coastal state in Emilia-Romagna, Arpae 2018). |
| **Line 210:** no dune slope analyses were included in the methodology nor the results so this information here is new, and makes everything confusing. | The mentioning of slope change will be removed since no specific quantitative analysis was performed for slopes - only visual assessment using the profiles in Figure 6b. |
| **Line 214:** the driftwood and debris reached the dune because the waves transported them and reached that area? Or were they blown by the wind? If it was the waves (in the 2021 orthomosaic it looks like the watermark is really close to the dune), maybe is worth stating that those fences are likely to be washed away and/or destroyed over time, maybe with the impact of storms, jeopardizing the restoration of the dunes. Please check. | The driftwood and debris were transported by waves. The suggestion to include the possibility of the fences to be destroyed over time will be reflected in the revised manuscript. |
| **Line 217:** I am not sure what you mean, did Puijenbroek et al (2016) study the same | The study of van Puijenbroek et al. (2016) was about the embryo dune development in |

| | |
|---|---|
| dune? | the Netherlands.

What we were trying to express in this part is that we observed a similar result to that of their study, wherein increase in vegetation colonization has contributed to the stabilization of sand accumulation within the dunes.

We will revise the flow of discussion accordingly. |
| **Line 224:** How were these fences installed? What was the geometry, orientation, etc? It would be good to know which fence configuration worked and which one didn't. | The selection of the most suitable nature-based solution intervention was done after a geographic, environmental, lithological, hydrogeological, geomorphological and hydrodynamical characterization of the study area. All data were collected during the three-year project that is mentioned in the text (Lines 89-97)

These are the specific details of the fence configuration:

A grid of windbreak fences (ganivelles, Figure 2b in the manuscript) was set parallel to the coast, held together by galvanized and twisted iron wire. The first line was positioned at the dune foot; the second line is about 2 m from the first line towards the sea; the two lines are connected by perpendicular portions, every 8 m ('mesh' technique). The intervention stretches across 465 m.

The ganivelles are fences made of chestnut (chosen for its resistance), whose purpose is to block the wind loaded with sand and consequently favor its accumulation to recreate the dune. They are therefore used to favor the creation of embryo dunes; to reduce the erosive effect of the wind and in general in the discontinuities of the dune system; to prevent the transport of sand towards the inland, sediment that would be definitively lost from the coastal sedimentary cell; and to prevent access and trampling on the dune.

The spacing or distance between stakes, also called permeability to wind flow, varies from |

| | 30 to 100 mm; the most common permeabilities in dune reconstruction interventions are 60-65 mm, 75-80 mm. In our case the stakes were 10 cm spaced. The most common heights are 1.00m / 1.20m. In our case, the initial stake/ fence height was 1.20m and the poles where they were fixed was 1.80m. |
|---|---|
| **Line 241:** you should mention which fence configuration worked | The fence configuration used in the study area will be mentioned in this line. The study of Hanley et al. (2013) will be included as supporting literature http://dx.doi.org/10.1016/j.coastaleng.2013.10.020 |
| **Line 249:** errors could also be the result of the GCPs configuration and number used, but this was not explained in the methods. | The GCP configuration (number of GCPs, location in the study area, geographic coordinates applied, which ones were selected as control and check points) will be included in the methods. |
| **Conclusions:**

**Line 271:** which decrease of blowout features? There is no description of the blowouts before, how many blowouts were in the area? The reader cannot reach to a conclusion related to something that has not been correctly introduced before. Please check. Also, do you mean that the blowouts were naturally sealed by vegetation growth promoted by the fence installation? How long did it take for the blowouts to be sealed? This could be interesting to know. | Information about blowouts and their presence in the study area will be included in the revised introduction and methods. An additional figure will be produced to support the conclusion provided in Line 271. |

---

## Author Response (AR1)

| Reviewer 1 | Authors |
|---|---|
| **GENERAL COMMENT:** In this manuscript, drones and GPS were used in two surveys (2016 and 2021) to assess the efficiency of using fences to restore the Bevano dune ridge (Ravenna, Italy). The DEMs and orthomosaics obtained were used to compute a DEM of difference and to classify the area in 3 classes, respectively. The previous allowed the analysis of dune geomorphic and vegetation changes over time. At its present form the manuscript is not ready to be published, and I suggest the authors to perform major changes. The intention of this review is to help them to improve the manuscript, and I hope they find the comments constructive. The different sections are not well connected, as the authors do not introduce properly some aspects (e.g. blowouts, dune slopes) neither in the introduction nor the methodology, but then discuss about those all of a sudden. In addition, important aspects of the methodology (e.g. drone flight elevation, number of control points and configuration used, or image classification accuracy assessment) were not performed, or at least not explained in the text. In addition, the result and discussion sections are a bit vague (see specific comments below). On the other hand, figures should be carefully revised and improved, mainly increasing the size of the font used as they are hard to read, adding the correct units, and adequate size bars and scales. I also recommend the authors to re-phrase some paragraphs and make them easier to read and follow, avoid repetitions, and to review the English, as there are typos throughout the text. | Revisions have been implemented in the figures and the rest of the manuscript to include aspects that have been missed in the initial version. Drone flight survey information (flying height, number of control points, and configuration) is in Lines 115-122.

Updated information regarding the image classification used for the cover analysis (Lines 160-170).

All the specific comments per section were considered and were reflected in the updated version. |

| Specific comments | |
|---|---|
| **Abstract:**

Lines 8-10: remove that phrase since it would be better placed at the introduction, it does not say much at the abstract. | The phrase has been removed from the abstract and have been integrated in the introduction (Lines 34-35). |
| Line 11: what's the temporal scale of your study? It is not mentioned but it is important to indicate it, as you later on talk about dune progradation of 3-5 metres. Also, how many drone surveys were performed? | The temporal scale is 5 years using two surveys (October 2016 and October 2021). This information has been stated in the introduction and has been included in the abstract (Line 11). |
| Line 14: Do the fences really prevent dune erosion? Or instead they promote dune recovery and growth? | Revised as suggested: dune fences prove to be effective in promoting dune recovery and growth (Lines 12 – 14). |
| Lines 14-16: I suggest the authors to include numbers when mentioning the results: how much sand accumulation was measured and how the vegetation cover changed, instead of mentioning 'increase in vegetation' or 'significant sand accumulation'. | Quantities were included (Lines 12 – 18). |
| Line 19: I would remove the line about the GCD toolset. | This line has been removed. |
| **Introduction:**

 Lines 29-30: Check the order of some references in the text, I'm not sure they are placed correctly. | Revised (Lines 29-31). |
|  Line 34: Not sure these are the best | These references were removed. |

| | |
|---|---|
| references. | |
| Line 36: For consistency, it is better to use only one term. Before the authors were using 'UAV' and now 'UAS' | The term UAV has been used throughout the manuscript. |
| Lines 45-49: it is a bit repetitive | These lines have been revised and additional literature provided in the Line 49 comment has been included (Lines 42 – 48). |
| Line 49: There are already some studies in the Emilia-Romagna region, please check the paper of Duo et al (2021): https://doi.org/10.3390/rs13091823 | Revised (Lines 42 – 48). |
| Figure 1: The legends are hard to read. | The legends in Figure 1 have been revised. Note: all figures were revised accordingly. |
| Lines 92-95: there is no mention about the characteristics of the blowouts in the area, which are mentioned in the abstract but not here. | Information about the blowouts characteristics in the area has been added (see Figure 10 and Line 95). |
| Figure 2: Legends and numbers are really small. Also, what are the red dots (A, B, and C) indicating? | Figure 2 has been revised to improve the readability of the legends. Points A, B, and C represent areas in the back dune, foredune, and beach along a section of the project. This information will be included in the figure caption (Line 105). Note: all figures were revised |
| Line 101: how frequent were the drone surveys? | Annual drone surveys for monitoring were done after the installation in 2016. In this paper, we do not analyze the yearly surveys but only the years 2016 and 2021. Line 101 revised to: The availability of annual UAV topographic surveys after the fence installation in 2016 and the availability of open-source tools can address the gaps in |

| | quantifying the restoration efficacy (Lines 106-107). |
|---|---|
| Line 102: qualitative data analysis? | This should have been "quantitative" instead of "qualitative" data analysis. Revised (Line 108). |
| **Materials and methods:**

Lines 114-116: which UAV software? what was the flight elevation? 70% of front and side overlap? How were the targets distributed in the area and how many did you use? In figure 3 targets are referred to as GCPs, use the same term for consistency. | The software used for the drone flight planning and execution was Pix4DCapture. Flying heights that were used in the survey are both 20m. Upon verifying the project files, the side and front overlap is at 80% instead of 70%. The term GCPs has been uniformly applied. Revised (Lines 118 – 128). |
| Lines 120-121: not important, I would remove these lines | Lines removed. |
| Line 125: how many points did you use? Check points are also used to assess the horizontal and vertical accuracy of the 3D models, how did you assess those? These lines are a bit hard to understand, please re-phrase. Also, how did you remove the vegetation from the DEM?

Lines 125-134: I think there is too much info that is not needed here | For the 2016 data, a total of 13 GCPs were utilized. 9 were considered control points and 4 GCPs were checkpoints. The 2021 data has a total of 15 GCPs - 10 were controls, 5 were checkpoints.

The horizontal accuracy was assessed using the RMSE values of the check points (Lines 133-134). Vertical accuracy was evaluated by comparing the values with the GPS profiles carried out in the field (Lines 134-137)

The vegetation from the DEM was removed using the automatic ground point classification (using the default parameters) of the Agisoft Metashape software. Basically, non-ground points were considered as vegetation and other structures that are present in the area. Only the points that were classified as ground were utilized to create the concurrent DEMs (Lines 129-133) |
| Line 137: ArcGIS 10.x? | Revised to the software version used, which |

| | |
|---|---|
| | is ArcMap 10.8.2 (Line 131) |
| Lines 125-163: I suggest to simplify and re-phrase these lines, they are a bit confusing and hard to follow as they are right now. | Revised (Lines 129 – 172). |
| Line 166: why 1 x 1 m of grid resolution? Also, how was the accuracy of the unsupervised classification assessed? This is important but it is not mentioned. | The method applied to detect the presence/absence of vegetation is based on a statistical approach similar to Silvestri et al. (2022) and is based on the visual inspection of the centroids of the grid cells: if the centroid falls on bare soil we have absence of vegetation while if it falls on a vegetated pixel of the photo, we have presence of vegetation. As the orthophotos have a resolution of 0.1 m x 0.1 m, a 1 x 1 m grid resolution allows us to sample one pixel (corresponding to the centroid) every 100 pixels included in each grid cell. This method is similar to a classic visual ecological survey performed in the field with 0.1m x 0.1m plots placed at a distance of 1m from each other along a transect, but in this case it is performed on an orthophoto instead, with the assumption that the operator has a clear overview of the area and can clearly distinguish between vegetated and non-vegetated (either with bare sand or logs/debris) pixels. |
| | Therefore, the accuracy of the method depends on the ability of the operator as well as on the image quality. In the new version of the paper, we have included these details about the method and stated the assumptions. As for the classification, we decided not to apply any classification method because no specific field surveys were originally designed to collect ground truths on vegetation, vanishing any possibility of validating the results unless visual interpretation were used to perform such validation, with an approach similar to the one that we applied directly to the orthophotos in our statistics. |
| | Revised (Lines 160 – 173) |

| | |
|---|---|
| **Results:**

Line 175: Maybe explain where those values are located, in which profiles? help the reader a bit to understand what happened in your area of study, and especially what happened with the dune (do not forget it is the goal of your work) | Revised (Lines 177 – 181).
Note that this part is focused on the DEM development and validation process. The dune changes were discussed in the latter part (3.2 Geomorphic and vegetation cover changes). |
| Figure 5: profile 10 is not included. | Profile 10 has been included in the revised Figures 5 and 6. |
| Line 184: if no vertical accuracy of the 2016 DEM could be obtained, which error associated to this DEM did you propagate to obtain the DoD? only the error of the 2021 DEM? | The propagated error in the 2016 DEM includes the total RMSE of the selected control points (Lines 145 – 146). |
| Line 185-193: These lines are a bit repetitive and confusing. It could help if you describe in which areas of the study site you found the erosion or the deposition of sand, using the map of Figure 6. Was erosion identified at the dune or the beach? Or both? Again, these results should be focused on describing the changes on the dune especially, as it is the 'goal' of your work. | Revised (Lines 189 – 197). |
| Line 186: use erosion and deposition instead of lowering and raising, also in the figures | The authors used erosion and deposition in the revised version. |
| Figure 6: the numbers and letters are very small. Also, use metres instead of km for the distance bar at the map (also in the rest of the figures). | The GCD map was enlarged, and the scale bar was changed to meters. Profile 10 was also included in the revised figure.

All map scale bars were revised from kilometers to meters. |

| | |
|---|---|
| Figure 7: please, check carefully the units of each variable, in all the figures and the text. And also the scale of the Y axis, why the vertical scale is not consistent along the negative and positive values in the elevation graph? | Thank you for noticing the error. We have already changed the units for the areal, volumetric, and average depth of changes. We have also corrected the vertical scale and term of the average depth change (elevation). Aside from correcting the vertical scale, both surface erosion and deposition are represented by positive values to allow easier comparison. The terms lowering and raising were also revised to erosion and deposition. (Lines 200 - 201). |
| I am missing some measurements here about the dune progradation that were mentioned in the abstract? | Progradation result was indicated in Line 221-222 (Section 4.1) |
| Line 201: what happens with profile 10? It is not included in the previous analysis but it is included in Figures 8 and 9, but no results associated to it are explained. It looks like vegetation cover did not increase that much compared to the debris or logs. At this point, I wonder whether the unsupervised classification method differentiates well both classes (logs or debris from vegetation)? Did you assess the classification accuracy?How did you do it? In the methods is not explained and neither is here, and it is quite important | Profile 10 was already included in Figures 5 and 6. This profile has been processed but was not included in the previous manuscript due to layout restraint. As mentioned in the comment Line 166, the statistical approach used to assess the vegetation cover is based on the visual inspection of the 2016 and 2021 orthomosaic images. Additional information pertinent to the cover analysis method has been included (Lines 202 – 208) |
| Figure 8: the 2021 dune crest and shoreline are shown over the 2016 orthomosaic, please be careful. Why are not the rectangles perpendicular to the shoreline? The transition between blue and green is not easy to see, maybe change the colours. | The 2021 data was only used as visual reference for the dune crest and shoreline. The rectangles are not perpendicular since the boundary condition used in the fishnet creation was the profile transects. We opted not to adjust the default fishnet output since the goal is to observe cover change between the 2 survey periods. The maps were improved in terms of color, font size, scale bar, and legend details. |

**Discussion:**

| | |
|---|---|
| **Line 208**: significant compared to what? are any other similar areas experiencing less deposition for the given time span of 5 years? You could compare accretion values and progradation rates measured in other similar areas, in order to know if is significant or not. Maybe it would be good to explain what's the sediment budget in the area, is there enough sediment arriving or the sediment is scarce there? | The sediment budgets along the Emilia-Romagna coast are very complex and site specific. It is difficult to compare the sediment budget of a nearby area to the study area. We have included additional information based on the last published report of the regional agency for environment (Report of the coastal state in Emilia-Romagna, Arpae 2018). Revised (Lines 217 – 219). |
| **Line 210:** no dune slope analyses were included in the methodology nor the results so this information here is new, and makes everything confusing. | The mentioning of slope change will be removed since no specific quantitative analysis was performed for slopes - only visual assessment using the profiles in Figure 6b. |
| **Line 214:** the driftwood and debris reached the dune because the waves transported them and reached that area? Or were they blown by the wind? If it was the waves (in the 2021 orthomosaic it looks like the watermark is really close to the dune), maybe is worth stating that those fences are likely to be washed away and/or destroyed over time, maybe with the impact of storms, jeopardizing the restoration of the dunes. Please check. | The driftwood and debris were transported by waves. The suggestion to include the possibility of the fences to be destroyed over time will be reflected in the revised manuscript. Revised (Lines 254 – 256, 259-260) |
| **Line 217:** I am not sure what you mean, did Puijenbroek et al (2016) study the same dune? | The study of van Puijenbroek et al. (2017) was about the embryo dune development in the Netherlands.

What we were trying to express in this part is that we observed a similar result to that of their study, wherein increase in vegetation colonization has contributed to the stabilization of sand accumulation within the dunes. Revised (Lines 228 – 229).

Publication date of |

| | |
|---|---|
| **Line 224:** How were these fences installed? What was the geometry, orientation, etc? It would be good to know which fence configuration worked and which one didn't. | The selection of the most suitable nature-based solution intervention was done after a geographic, environmental, lithological, hydrogeological, geomorphological and hydrodynamical characterization of the study area. All data were collected during the three-year project that is mentioned in the text (Lines 87-95) |
| | These are the specific details of the fence configuration: |
| | A grid of windbreak fences (ganivelles, Figure 2b in the manuscript) was set parallel to the coast, held together by galvanized and twisted iron wire. The first line was positioned at the dune foot; the second line is about 2 m from the first line towards the sea; the two lines are connected by perpendicular portions, every 8 m ('mesh' technique). The intervention stretches across 465 m. |
| | The ganivelles are fences made of chestnut (chosen for its resistance), whose purpose is to block the wind loaded with sand and consequently favor its accumulation to recreate the dune. They are therefore used to favor the creation of embryo dunes; to reduce the erosive effect of the wind and in general in the discontinuities of the dune system; to prevent the transport of sand towards the inland, sediment that would be definitively lost from the coastal sedimentary cell; and to prevent access and trampling on the dune. |
| | The spacing or distance between stakes, also called permeability to wind flow, varies from 30 to 100 mm; the most common permeabilities in dune reconstruction interventions are 60-65 mm, 75-80 mm. In our case the stakes were 10 cm spaced. The most common heights are 1.00m / 1.20m. In our case, the initial stake/ fence height was 1.20m and the poles where they were fixed was 1.80m. |
| | Revised (Lines 87 – 102). |

| | |
|---|---|
| **Line 241:** you should mention which fence configuration worked | The fence configuration used in the study area has been mentioned. Additional literature by Hanley et al. (2013) has been included in the introduction wherein the dune fence was based on (Lines 100 – 103).

This line has also been revised in the discussion (Lines 253 – 254). |
| **Line 249:** errors could also be the result of the GCPs configuration and number used, but this was not explained in the methods. | The GCP configuration (number of GCPs, location in the study area, geographic coordinates applied, which ones were selected as control and check points) has been included in the methods.

Revised (Lines 272 – 274). |
| **Conclusions:**

**Line 271:** which decrease of blowout features? There is no description of the blowouts before, how many blowouts were in the area? The reader cannot reach to a conclusion related to something that has not been correctly introduced before. Please check. Also, do you mean that the blowouts were naturally sealed by vegetation growth promoted by the fence installation? How long did it take for the blowouts to be sealed? This could be interesting to know. | Information about blowouts and their presence in the study area have been included in the revised introduction and methods (Line 95; Lines 206 – 207)

An additional figure was created to support the conclusion (Figure 10 in Line 214) |
| **Reviewer 2** | **Authors** |
| Improve the quality of the figures since a few looks faded out.

The flow of the main text could be improved. However, the current version is acceptable. | Thank you very much for the feedback. So far, all the figures have been improved and are reflected in the revised manuscript.

The overall flow was revised, with all suggestions considered. |

| Community comment 1 | Authors |
|---|---|
| The summary presents a study of the recovery of foredunes through the fence installation method.

The methodology used is adequate and the results show recovery in a large sector of the project area. | Thank you very much for the positive feedback. Major revisions have been implemented in the final manuscript. |
| **Other relevant changes** | |
| Old manus:

1. Lines 59 – 61 (Introduction)
2. Section header 2.1 UAV survey and elaboration (Line 108)
3. Line 271 (conclusion)
4. van Puijenbroek et al. (2016) | New manus:

1. Revised (Lines 58 – 59)
2. Section header was removed.
3. Revised (Lines 287 – 288)
4. Revised to van Puijenbroek et al. (2017); upon checking the reference list, date of publication should be 2017 instead of 2016. |

---

## Referee Report (RR1)

I find this work to be of good quality. Every section is written scientifically. I endorse this work for publication.

I propose a minor enhancement: the use of contrasting colors in the figures and making legend symbols bigger to make them clearly visible.

Minor suggestions:

Line 39: what is the point Digital Elevation Models (DEMs)? I think "point" should be removed.

Line 135: How the GPS profiles are better than collecting point data and then assessing vertical accuracy from them? It may be mentioned somewhere in the methodology.

Figure 4: It is suggested to use contrasting colors for the restoration fence. It is hardly visible.

Figure 8: Cover type legend colors are not clearly visible. Make symbols a little bigger.

---

## Author Response (AR2)

| Anonymous Referee #3 | Authors |
|---|---|
| **GENERAL COMMENT:** I find this work to be of good quality. Every section is written scientifically. I endorse this work for publication.

I propose a minor enhancement: the use of contrasting colors in the figures and making legend symbols bigger to make them clearly visible. | Thank you very much for taking the time to review the manuscript.

All figures with maps (Figures 2, 4, 6, 8, and 10) were revised to improve the display and readability of the features and the legends. |
| *Specific comments* | |
| Line 39: what is the point Digital Elevation Models (DEMs)? I think "point" should be removed. | The word "point" has been removed. |
| Line 135: How the GPS profiles are better than collecting point data and then assessing vertical accuracy from them? It may be mentioned somewhere in the methodology. | This line was revised to clarify how the DEM vertical accuracy was assessed (Lines 135 -138)
New line:
"… **the vertical accuracy was evaluated by comparing the DEM values extracted using QGIS Profile Tool plugin version 4.2.0 to the GPS point values. The computed DEM points and the measured GPS points along each transect were visualized by scatterplot in Python 3.9.**"

Info about GPS data collection (Lines 117-118) was also improved to give clearer context as to how it was performed.

Revised: "Collected datapoints include several profiles across the beach from the coastline to the back dune **with a measuring interval of one (1) meter space**." |
| Figure 4: It is suggested to use contrasting colors for the restoration fence. It is hardly visible. | Figure 4 has been revised accordingly. |
| Figure 8: Cover type legend colors are not clearly visible. Make symbols a little bigger. | Figure 8 has been revised accordingly. |

| | |
|---|---|
| *Other relevant changes:*

Old manus:

  1.  Validation on the sample transects (Figure 4b) was performed using regression analysis shown in Figure 5 (Line 177)
  2.  the coastal zones of the Emilia-Romagna (Line 293). | New manus:

  1.  Validation on the transects (Figure 4b) was performed using linear regression shown in Figure 5. (Line 177)
  2.  the coastal zones of Emilia-Romagna (Line 293). |